# Velocity Time Integral: A Novel Method for Assessing Fetal Anemia

**DOI:** 10.3390/children10071090

**Published:** 2023-06-21

**Authors:** Ettie Piura, Offra Engel, Neta Doctory, Nisim Arbib, Tal Biron-Shental, Michal Kovo, Shmuel Arnon, Ofer Markovitch

**Affiliations:** 1Department of Obstetrics and Gynecology, Meir Medical Center, Kfar Saba 4428164, Israel; ettiepiura@gmail.com (E.P.); offra.engel@gmail.com (O.E.); narbib@gmail.com (N.A.); tal.biron-shental@clalit.org.il (T.B.-S.); michal.kovo@clalit.org.il (M.K.); markovitch.ofer@clalit.org.il (O.M.); 2Sackler School of Medicine, Tel Aviv University, Tel Aviv 6997801, Israel; neta.doctory@gmail.com; 3Department of Neonatology, Meir Medical Center, Kfar Saba 4428164, Israel

**Keywords:** peak systolic velocity, Doppler, velocity time integral, fetal anemia, stroke distance, middle cerebral artery, umbilical artery

## Abstract

The velocity time integral (VTI) is a clinical Doppler ultrasound measurement of blood flow, measured by the area under the wave curve and equivalent to the distance traveled by the blood. This retrospective study assessed the middle cerebral artery (MCA) VTI of fetuses in pregnancies complicated by maternal alloimmunization. Doppler indices of the MCA were retrieved from electronic medical records. Systolic deceleration-diastolic time, systolic acceleration time, VTI, and peak systolic velocity (PSV) were measured at 16–40 weeks gestation. Cases with PSV indicating fetal anemia (cutoff 1.5 MoM) and normal PSV were compared. The study included 255 Doppler ultrasound examinations. Of these, 41 were at 16–24 weeks (group A), 100 were at 25–32 weeks (group B), and 114 were at 33–40 weeks (group C). VTI increased throughout gestation (5.5 cm, 8.6 cm, and 12.1 cm in groups A, B, and C, respectively, *p* = 0.003). VTI was higher in waveforms calculated to have MCA-PSV ≥ 1.5 MoM compared to those with MCA-PSV < 1.5 MoM (9.1 cm vs. 14.1 cm, respectively, *p* < 0.001), as was VTI/s (22.04 cm/s vs. 33.75 cm/s, respectively; *p* < 0.001). The results indicate that the MCA VTI increases significantly among fetuses with suspected anemia, indicating higher perfusion of hemodiluted blood to the brain. This feasible measurement might provide a novel additional marker for the development of fetal anemia.

## 1. Introduction

Doppler velocimetry of the fetal peak systolic velocity (PSV) of the middle cerebral artery (MCA) is a non-invasive test that is used to identify fetuses at risk for anemia [1,2,3,4,5].

An MCA PSV above 1.5 multiples of the median (MoM) indicates moderate-to-severe fetal anemia with high sensitivity and low false-positive rates [2]. This noninvasive method is the most accepted tool for predicting fetal anemia in pregnancies complicated by maternal alloimmunization [2,3].

PSV represents the maximum blood flow velocity. However, this measurement does not express other components of the flow or the distribution of blood velocity throughout the duration of each heartbeat.

The velocity time integral (VTI) is a clinical Doppler ultrasound measurement of blood flow, equivalent to the area under the curve, which indicates the sum of all individual velocities throughout the heartbeat period. (Figure 1). This index represents the distance traveled by the blood and is an indicator used to predict intravascular blood volume. It can be referred to as stroke distance since it indicates how far blood travels during the flow period. While the PSV Doppler index expresses only the maximum systolic flow rate, the VTI represents the distance blood flows during one heartbeat cycle in the vessel measured and, therefore, is an important indicator of tissue perfusion [6].

VTI aids in assessing changes in stroke volume and is useful for assessing changes in cardiac output. The clinical use of the VTI component is well-known and widely studied in adult cardiology [7,8,9].

In contrast to the utility of VTI assessment in various postnatal clinical conditions, it is utilized much less often in fetuses. Most prenatal studies of the VTI index include central blood vessels in fetuses with congenital heart malformations [10,11,12]. The utility and significance of measuring the VTI of blood flow in fetal peripheral vessels have not been reported in depth.

While the MCA-PSV expresses the maximum rate only, the VTI depicts additional anemia-induced hemodynamic changes in tissue perfusion.

The purpose of the current study was to describe the fetal MCA VTI in pregnancies complicated with maternal alloimmunization and to compare the values between those not suspected of having developed fetal anemia and those suspected of moderate-to-severe anemia. To date, this is the first study of VTI in the MCA of fetuses at risk of developing anemia.

The hypothesis was that the fetal MCA-VTI would increase in parallel to the well-known increase in the maximum blood flow velocity throughout gestation, and that in fetuses suspected of having moderate to severe anemia, the MCA-VTI would increase even more.

## 2. Materials and Methods

This retrospective study was performed at the Obstetrics and Gynecology Ultrasound Unit in a tertiary academic center. Pregnant patients at risk for fetal anemia due to maternal alloimmunization and referred for Doppler ultrasound MCA-PSV measurement from January 2015 to May 2021 were included. Data regarding obstetric parameters and Doppler indices were retrieved from the electronic medical records.

The study included Doppler ultrasound examinations of the MCA PSV performed at 16 to 40 weeks of gestation in pregnancies complicated by maternal alloimmunization. Women with high titers of the following antibodies were included: anti-D, anti-S, anti-M, anti-C, anti-c, anti-Kell, anti-E, anti-e, anti-FYA, anti-CW, anti-JK, and anti-P1 (>1:4 or >1:1 for anti-Kell).

The Doppler results of 3 groups based on the gestational week of the measurements were compared. Group A included measurements performed between 16 and 24 weeks of gestation; Group B included measurements between 25 and 32 weeks; and Group C included measurements performed between 33 and 40 weeks.

Exclusion criteria were multiple gestations, fetal anomalies, genetic aberrations, and failure of Doppler wave recording.

Doppler ultrasound of fetal MCA velocity was performed using Voluson E8 and E10 ultrasound machines (GE Healthcare Ultrasound, Milwaukee, WI, USA) with an abdominal 3.5 MHz curved-array transducer. The MCA velocity was demonstrated, measured, and recorded at least twice during each examination to assess reproducibility.

The MCA was imaged by color flow Doppler. Pulsed Doppler evaluation of the MCA was performed on the MCA branch proximal to the transducer, close to the maternal skin. The Doppler pulsed sample volume gate was placed over the vessel just where it bifurcates from the carotid siphon at the Circle of Willis. The sample volume gate included the entire internal diameter of the MCA at the point of measurement. Measurements were taken with the fetal head in the transverse position during a period of apnea or no fetal movement to reduce the chance of compensatory fetal heart rate acceleration causing a false elevation in the peak systolic velocity. All examinations were performed by an obstetric ultrasound technician or a sonographer specializing in obstetrics and gynecology.

Doppler recordings were obtained according to the department’s PSV measurement protocol, with the angle of insonation as parallel to the direction of flow as possible and not higher than 30 degrees. Each measurement was performed at least twice for reproducibility.

The MCA-PSV was reported in the patient’s medical record as a function of gestational age or as multiples of the median (MOM).

All studies were digitally acquired and stored using a PACS digital imaging workstation. The following indices were recorded from the MCA spectral tracing: PSV, HR, resistance index (RI), pulsatility index (PI), and systole/diastole ratios (S/D). In addition, the area within the spectral wave was measured by curve tracing [Figure 1]. The area under the curve tracing, referred to as the VTI, was calculated automatically by the PACS digital imaging program. The curve was traced on the same waveform used for the PSV measurement. The MCA-PSV MoM was calculated using an online PSV MoM calculator (Perinatology.com by MARK Curran).

### 2.1. Study Measurements

The following measurements were calculated (Figure 1). The velocity time integral (VTI), defined as the distance the blood flowed in each heartbeat, was calculated by the area under the curve, representing the waveform integral.

Velocity time integral per second (VTI/sec) was defined as the distance blood flowed in each second, calculated by the area under the curve of one heartbeat multiplied by the heartrate divided by 60. The stroke distance was calculated as the VTI multiplied by the heart rate. Systolic acceleration time (SAT) was measured as the time from the beginning of beat systole to the PSV. Systolic deceleration-diastolic time (SD-DT) was measured as the time from the PSV to the end of the beat. Each waveform length was measured, defining the duration of one heartbeat. Systolic acceleration time to systolic deceleration-diastolic time ratio (SD-DT/SAT) was defined as the ratio between the time from PSV to the end of the beat and the time from the beginning of the beat systole and PSV.

### 2.2. Statistical Analysis

The data were described as the mean and standard deviation. Differences between the two groups were evaluated using a *t*-test and between the 3 groups with a one-way ANOVA. Continuous variables, such as gestational age at examination and MCA-PSV MoM data, were compared using Pearson correlations. *p* < 0.05 was considered statistically significant. All data were analyzed using SPSS-26 (IBM Corp., Armonk, NY, USA).

## 3. Results

A total of 255 MCA Doppler velocimetry examinations of 56 pregnancies complicated by maternal alloimmunization were included in this study. Examinations were performed throughout 16–40 weeks of pregnancy, with 41 examinations at 16–24 weeks (group A), 100 at 25–32 weeks (group B), and 114 examinations at 33–40 weeks of pregnancy (group C).

The waveform velocimetry indicators of the study population are presented in Table 1. The mean MCA-PSV increased significantly as the pregnancy progressed (28.5 cm/s in group A, 47.6 cm/s in group B, and 60.8 cm/s in group C, *p* < 0.001), with no significant difference between the groups in MCA-PSV MoM (1.1, 1.2, and 1.2 for groups A, B, and C, respectively).

The mean heart rate decreased significantly in later gestation (147.1 bpm, 141.7 bpm, and 140.6 bpm in groups A, B, and C, respectively, *p* = 0.015), while the mean duration of each heartbeat increased (0.386 s, 0.414 s, and 0.415 s in groups A, B, and C, respectively, *p* = 0.01). However, the SD-DT/SAT ratio did not change significantly between gestational age groups (Table 1).

The VTI increased significantly throughout gestation (5.5 cm/s, 8.6 cm/s, and 12.1 cm/s in groups A, B, and C, respectively, *p* = 0.003). VTI per sec (calculated by VTI × HR/60) was also significantly higher in later gestation (13.4, 20.2, and 28.4 in groups A, B, and C, respectively) (*p* < 0.0001; Figure 2).

Among 17/255 (6.7%) Doppler velocimetry examinations, the MCA-PSV MOM was ≥1.5, strongly suggestive of moderate-to-severe anemia. All 17 examinations were performed at 28 + 3 to 38 + 6 weeks of gestation, with 12 examinations in group B (gestational weeks 25–32) and 5 in group C (gestational weeks ≥ 33; Table 2).

The VTI was significantly higher in waveforms calculated to have MCA-PSV ≥ 1.5 MoM in comparison to those with MCA-PSV < 1.5 MoM (9.35 cm vs. 14.1 cm, respectively, *p* < 0.001), as was the VTI per sec (22.04 cm/s vs. 33.75 cm/s, respectively; *p* < 0.001; Figure 3).

The mean heart rate, mean time of each heartbeat, SAT, SD-DT, and the ratio between them, did not change significantly between cases with MCA-PSV ≥ 1.5 MoM and those with MCA-PSV < 1.5 MoM (Table 2).

The pregnancies in this study were managed based on clinical features and fetal MCA PSV MOM levels. Of the 17 cases with fetal PSV MOM ≥ 1.5, 2 underwent intrauterine transfusion. Labor was induced in 10 cases (6 due to late third-trimester gestational age, 3 due to non-reassuring fetal heart rate monitoring, and 1 following the development of eclampsia). An additional patient experienced spontaneous labor before intervention. Four patients received further treatment at another facility but were lost to follow-up.

## 4. Discussion

The current study examined the change in VTI of the fetal MCA in pregnancies complicated by maternal alloimmunization. The results showed increases in the distance blood flows during each heartbeat and in each second, expressed by the waveform VTI and the VTI per sec, throughout pregnancy. This increase in flow with gestational age agrees with the known physiologic increase in the PSV throughout pregnancy, suggesting vascular perfusion changed to accommodate the requirements of the growing fetal brain. In addition to the increases in MCA VTI throughout pregnancy, our results indicate further increases in MCA VTI values in fetuses with MCA-PSV ≥ 1.5 MoM compared to those without MCA-PSV levels suspected of moderate to severe anemia. The increases were seen even though most of the cases with MCA-PSV MoM ≥ 1.5 were in gestational age group B. This increase in the MCA stroke distance in fetuses suspected of having anemia indicates alterations in the fetal cerebral tissue perfusion induced by oxygen deprivation [6,10].

Doppler waveforms of fetal vasculature have been studied in various clinical conditions. Alterations in blood flow waveform indices are potential signs of fetal compromise or an expression of fetal deterioration. The use of MCA-PSV for diagnosing moderate-to-severe fetal anemia is a fundamental tool in the follow-up and management of fetuses who might develop anemia in conditions such as maternal alloimmunization [1,2,3,4,5,13]. This noninvasive tool is the most accepted option for predicting fetal anemia in pregnancies complicated by maternal alloimmunization [2,3].

However, increases in the MCA-PSV in anemic fetuses express only the maximum systolic flow rate and no other potential changes in cerebral blood flow waveforms. The demonstration of additional waveform alterations may contribute to our understanding of the risk of cerebral tissue damage induced by oxygen deprivation.

Previous studies found that fetuses with anemia have high cardiac output and decreased blood viscosity, resulting in high blood flow velocities. These findings were attributed to the “brain sparing effect” hypothesis, suggesting that decreases in the oxygen content of the blood delivered to the fetal brain (in cases of fetal anemia or intrauterine growth restriction) can induce cerebral vasodilatation [13,14]. MCA-PSV presents only a portion of the information, does not provide data regarding the entire waveform of the fetal MCA, and is not directly correlated to the cardiac output perfusion rate of the brain tissue.

The VTI represents the distance blood flows during one heartbeat cycle in the vessel measured. This distance affects the rate of tissue perfusion. VTI measurements for stroke volume calculation have been extensively discussed in adult cardiology studies.

The mitral valve VTI and the left ventricular outflow tract VTI have been suggested as useful for tracking changes in stroke velocity and cardiac output. The echocardiographic heart stroke volume is calculated by multiplying the VTI by the cross-sectional area of the blood vessel. Both the mitral valve VTI and the left ventricular outflow tract VTI have been used as parameters for assessing response to treatment such as fuid challenges, vasopressor therapy, inotropic support, or relief of obstructive shock mechanisms. As a general rule, a >15% increase in the VTI after a treatment indicates a concrete response to the therapy [15,16,17,18,19,20,21]. Therefore, the alterations in blood flow VTI are directly correlated to blood perfusion.

Moreover, the American Society of Echocardiography Guidelines and the Task Force of the European Society of Intensive Care Medicine Consensus both recommend using transesophageal or transtracheal echo for monitoring stroke velocity and cardiac output to determine response to medical and surgical therapies. Both the stroke velocity and the cardiac output calculations depend on VTI measurements. This index was found feasible and is frequently used in cardiology clinical practice. [15].

The utility of the VTI measurement is much lower in prenatal work-up than in postnatal healthcare and is rarely incorporated in obstetrical follow-up or in decision-making processes. The vascular response of the brain to acute asphyxia in sheep fetuses in late gestation was studied. After umbilical cord occlusion, there was an immediate increase in the VTI and in the VTI per minute [22]. Studies including human fetal blood vessel measurements of the VTI are almost exclusively limited to the central blood vessels and include fetuses with congenital heart malformations [10,11,12,23].

In a retrospective study of 27 fetuses diagnosed with left hypoplastic heart syndrome (HLHS), Mardy et al. examined the accuracy of the VTI of the fetal pulmonary veins in predicting which fetuses would need an emergent atrial septoplasty (EAS) after birth due to a restrictive atrial septum (RAS). They reported that the pulmonary vein prograde/retrograde VTI ratio was more accurate in predicting which neonates required EAS than was the fetal pulmonary artery pulsatility index [10]. These findings were supported by a study by Erik Michelfelder et al. on pulmonary venous VTI flow alternation in HLHS in association with RAS [11].

In a retrospective study, Wang et al. reported 69 fetuses with suspected coarctation of the aorta. The prenatal aortic isthmus Doppler indices of cases with (n = 31) and without (n = 38) aortic coarctation were compared. Their results showed a 25% improvement in identifying fetal aortic coarctation when the diastolic VTI/systolic VTI ratio was added to the basic model (the aortic isthmus Z SCORE) [12].

In another study, the PSV, end-diastolic velocity, pulsatility index, and VTI of the two segments of the posterior cerebral artery (PCA) were compared in fetuses with transposition of the great arteries and normal fetuses. The abnormality rate between the first segment of the PCA (PCAS1), the second segment of the posterior cerebral artery (PCAS2), and the MCA was compared. The MCA-VTI, PCAS1-VTI, and PCAS2-VTI were larger among the fetuses with transposition of the great arteries (all *p* < 0.05) [23].

While the MCA-PSV expresses the maximum flow rate only, the VTI is the area under the velocity spectral curve, which indicates the sum of all individual velocities throughout the ventricular ejection period. Therefore, VTI depicts additional anemia-induced alterations in vasculature that lead to increases in stroke distance.

The increase in cerebral blood flow associated with hemodilution is a well-characterized phenomenon caused by passive changes in rheology and active cerebral vasodilatation. Prenatal studies have mostly demonstrated an increase in velocity and a decrease in flow resistance indices as compensatory mechanisms [14]. However, there is a delicate balance between compensatory mechanisms promoting maintenance of cerebral blood flow and the potentially pathologic effect of increased perfusion of blood with decreased oxygen content [24].

In addition to being a sign of fetal anemia, the MCA VTI may represent a hemodynamic change in fetal brain circulation that has a potentially poor prognosis. Although the velocity of the blood flow in the MCA is increased, its viscosity is decreased; therefore, there is a relative decrease in red blood cells bearing oxygen.

The increase in the stroke distance of hemodiluted blood may be one of the factors leading to a higher risk of brain tissue injury among people with anemia [25,26,27,28,29]. If oxygen is reduced but substrate delivery is effectively maintained (i.e., pure or nearly pure hypoxia), the cells adapt in two ways. First, they can reduce non-obligatory energy consumption to some extent, initially by switching to lower energy-requiring states and then, as the insult becomes more severe, completely suppressing neuronal activity, which will cause neuronal depolarization above that threshold [29]. In a study by Ghi et al., four of seven fetuses with severe anemia were found to have various types of cerebral injuries, including intracerebral hemorrhage, periventricular leukomalacia, and ventriculomegaly. The cerebral injury was suspected to be the consequence of hypoxic encephalopathy due to decreased oxygen content in the fetal blood and an increase in hyperdynamic circulation of blood containing hypoxic molecules [28]. The increased cerebral stroke distance in the MCA of fetuses suspected of moderate to severe anemia demonstrated in this study is part of the hemodynamic changes in the fetal brain circulation in response to oxygen deprivation.

To our knowledge, this is the first study to describe the role of VTI measured in the MCA of fetuses in pregnancies complicated by maternal alloimmunization. The correlation between the peak blood flow velocity and the distance the blood flows in the MCA with each heartbeat has not yet been reported in fetal studies.

The results of this preliminary study serve as proof of concept for the MCA VTI index as an acceptable measure for predicting the risk of fetal anemia. The measurement of VTI is feasible and correlates with the fetal MCA-PSV in pregnancies complicated by maternal alloimmunization.

Our results, demonstrating an increase in the distance blood flows per second and especially in the oxygen-deprived blood of anemic fetuses, may contribute to the knowledge of prenatal compensatory mechanisms due to anemia and will promote the understanding of the mechanisms of potential tissue damage that occurs among anemic fetuses.

This study was limited by its retrospective nature. In addition, the incidence of maternal alloimmunization in pregnancy resulted in a relatively small sample size of fetuses with suspected moderate-to-severe anemia.

The role of MCA VTI as an additional marker for the development of fetal anemia and the early identification of fetuses with an increased stroke distance of hemodiluted blood who are at higher risk for brain tissue injury requires further study. An MCA VTI level nomogram according to gestational age is needed to define the pathological cut-off at each gestational stage. This study also indicates that the VTI might be further studied as a potentially important indicator for understanding and diagnosing alterations in fetal blood vessels and their reactions to various gestational conditions.

## Figures and Tables

**Figure 1 children-10-01090-f001:**
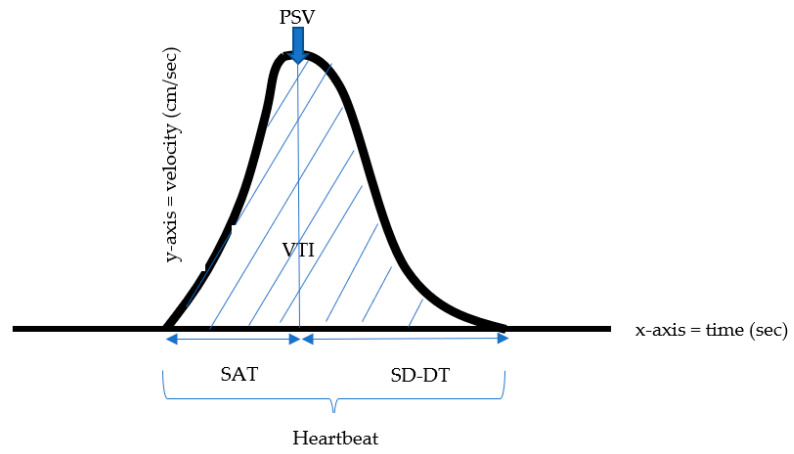
Schematic description of Doppler velocimetry. PSV (peak systolic velocity), VTI (velocity time integral), SAT (systolic acceleration time), SD-DT (systolic deceleration-diastolic time), HB (heartbeat). PSV, peak systolic velocity; VTI, velocity time integral; SAT, systolic acceleration time; SD-DT, systolic deceleration-diastolic time.

**Figure 2 children-10-01090-f002:**
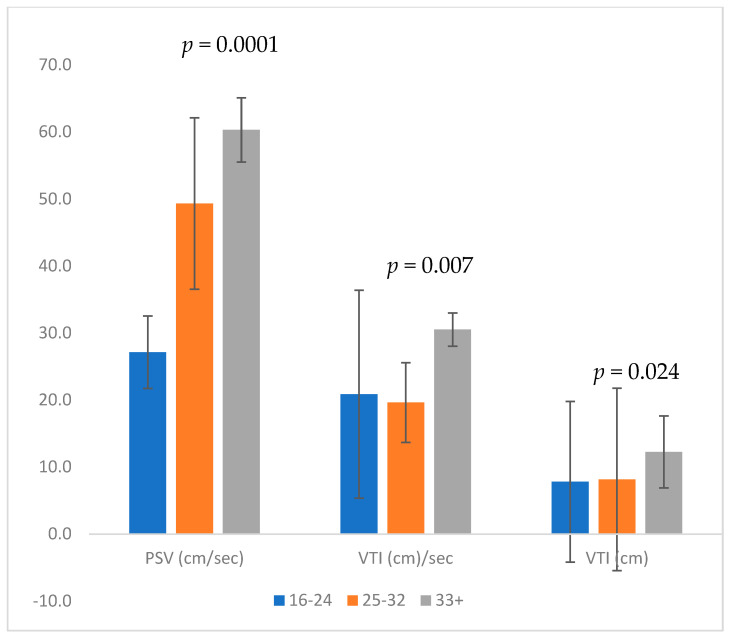
Measures of each gestational age group: 16–24 weeks, 25–32 weeks, and 33+ weeks. Middle cerebral artery peak systolic velocity (PSV, cm/s), middle cerebral artery velocity time integral per sec (VTI/s), middle cerebral artery velocity time integral (VTI, cm).

**Figure 3 children-10-01090-f003:**
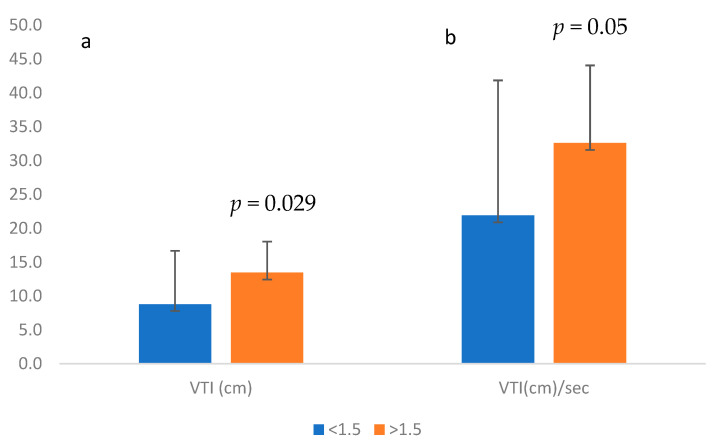
(**a**) MCA velocity time integral (VTI (cm) values when MCA peak systolic velocity (MCA PSV) was <1.5 or ≥1.5. (**b**) MCA velocity time integral per sec (VTI (cm/s)) when MCA peak systolic velocity (MCA PSV) was <1.5 or ≥1.5 MoM. MCA, middle cerebral artery; MoM, multiples of the median.

**Table 1 children-10-01090-t001:** Waveform velocimetry indicators by week of gestation.

Gestational Week	16–24	25–32	33+	*p*-Value
n	41	100	114	
Mean PSV (cm/s)	28.5	47.2	60.8	0.0001
Mean heartbeat (s)	0.386	0.414	0.415	0.01
Mean SD-DT (s)	0.3224	0.3463	0.3391	0.046
Mean SAT (s)	0.0724	0.068	0.0762	0.012
Mean SD-DT/SAT	5.048	5.596	4.862	0.06
Mean heart rate (beats per min)	147.1	141.7	140.6	0.015

PSV, peak systolic velocity; SD-DT, systolic deceleration-diastolic time; SAT, systolic acceleration time; SD-DT/SAT, systolic deceleration-diastolic time to systolic acceleration time ratio.

**Table 2 children-10-01090-t002:** Waveform velocimetry indicators by level of mid-cerebral artery peak systolic velocity.

Variable	MCA-PSV MoM < 1.5	MCA-PSV MoM ≥ 1.5	*p*-Value
n	237	17	
Mean PSV (cm/s)	48.82	72.09	<0.0001
Mean heartbeat (s)	0.41	0.39	0.16
Mean SD-DT (s)	0.34	0.32	0.11
Mean SAT (s)	0.07	0.07	0.8
Mean SD-DT/SAT	5.21	4.7	0.37

MCA, mid-cerebral artery; PSV, peak systolic velocity; MoM multiples of the median; SAT, systolic acceleration time; SD-DT, systolic deceleration-diastolic time; SD-DT/SAT, systolic deceleration-diastolic time to systolic acceleration time ratio.

## Data Availability

Restrictions apply to the availability of these data. Data were obtained from patient medical records and are available from the authors with the permission of the Meir Medical Center Ethics Committee.

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
