# Peer review of "Velocity Time Integral: A Novel Method for Assessing Fetal Anemia"

_children, 2023, doi:10.3390/children10071090_

Round 1

Reviewer 1 Report

The Authors present very interesting study. Manuscript is well presented and well performed. However, I have a few concerns:

1. I suggest adding the latest publications in this field to the bibliography

2. I suggest adding in the introduction how these two measurement methods differ (VTI vs MCA PSV)

Minor English editing required

Author Response

Reviewer 1: We thank the reviewer for his valuable remarks.

  1. Comment: “I suggest adding the latest publications in this field to the bibliography”

Response: The latest publications in the field of ultrasound surveillance and obstetric management of fetuses with suspected anemia were added to the paper as references 4, 5, 8 and 9.

  1. Comment: I suggest adding in the introduction how these two measurement methods differ (VTI vs MCA PSV)

Response: The difference between the PSV doppler index and the VTI doppler index for measuring velocimetry changes in fetal MCA were further explained in the Introduction.

“While the PSV doppler index expresses only the maximum systolic flow rate, The VTI represents the distance blood flows during one heartbeat cycle in the vessel measured and therefore, is an important indicator of tissue perfusion [6].”

Reviewer 2 Report

This article focuses on an important topic related to the clinical implications of sonographic evaluation regarding identifying fetal anemia and determining the optimal timing of intrauterine transfusion.

However, some suggestions could improve the quality of the article:

-       What are the sensitivity and specificity of the method, the accuracy, and the rates of false negative and positive results?

-       In how many fetuses was an intrauterine transfusion performed?

-       Doppler measurement of MCA-PSV, and estimation of fetal Hct drop or hemoglobin can be used to determine the timing of intrauterine transfusions in fetuses with erythrocyte alloimmunization. But can MCA-VTI be a more reliable tool?

-       Did the association of MCA-VTI with MCA-PSV increase the method's accuracy?

Kind regards

Some sentences need to be rephrased. In addition, corrections of agreement errors, incorrect spelling, and article misuse are required.

Author Response

Reviewer 2:

We thank the reviewer for his valuable remarks.

  1. Comment: what are the sensitivity and specificity of the method, the accuracy, and the rates of false negative and positive results?

  Comment: Doppler measurement of MCA-PSV, and estimation of fetal Hct drop or   hemoglobin can be used to determine the timing of intrauterine transfusions in fetuses with erythrocyte alloimmunization. But can MCA-VTI be a more reliable tool?

  Comment: Did the association of MCA-VTI with MCA-PSV increase the methods accuracy?

Response: This was a preliminary study that aimed to establish the MCA VTI (middle cerebral artery velocity time interval) index as a reliable measure for predicting the risk of fetal anemia and as a potential tool for studying cerebral hemodynamic changes caused by anemia. This study is the first to demonstrate the increase in MCA VTI levels throughout gestation, particularly in fetuses suspected of having moderate to severe anemia. However, it is important to note that the study focused on pregnancies at high risk for anemia due to maternal alloimmunization. The incidence of suspected anemia cases within this population was relatively low, which limited our ability to demonstrate the normal distribution of MCA VTI levels throughout gestation and to calculate the multiple of the median (MOM) of the MCA VTI at each week. As a result, the prediction and accuracy measurements of MCA VTI, as well as its combination with MCA PSV (peak systolic velocity), were beyond the scope of this preliminary study.

To address these limitations, we are currently conducting prospective studies involving a low-risk population of uncomplicated pregnancies. These studies will provide a clearer understanding of the normal VTI levels for each week of gestation, establish the normal MCA VTI MOM cutoff, and enable the calculation of the accuracy and predictive values for fetuses at risk of anemia.

  1. Comment: In how many fetuses was an intrauterine transfusion performed?

 Response: The pregnancies in this study were managed based on clinical features and fetal MCA PSV MOM levels. Of the 17 cases with fetal PSV MOM >1.5, 2 underwent intrauterine transfusion. Labor was induced in 10 cases (6 due to late third-trimester gestational age, 3 due to non-reassuring fetal heart rate monitoring, and 1 following the development of eclampsia). An additional patient experienced spontaneous labor before intervention. Four patients received further treatment at another medical facility and were lost to follow-up.

Round 2

Reviewer 2 Report

This article being a preliminary study, brings essential information related to the clinical implications of ultrasound evaluation in identifying fetal anemia; new research is needed to draw a series of conclusions.

Kind regards